# Determinants and strategies for environmental compliance in municipalities: Perspectives from KwaZulu-Natal Province, South Africa

Nqobile S. Zungu◉*, Gerhard P. Nortjé

Department of Environmental Sciences, Florida Science Campus, University of South Africa, Pretoria, Gauteng, South Africa

* 64069788@mylife.unisa.ac.za

## Abstract

### Background

A commitment to local environmental compliance is essential for ensuring a sustainable environment for future generations. Despite South Africa's extensive environmental laws, municipalities persistently exhibit non-compliance due to inadequate institutional capacity, political interference, and fragmented coordination.

### Methods

This study examines the determinants and strategies influencing environmental compliance in South African municipalities. Utilizing a qualitative research approach and the case study design, 10 municipalities within the KwaZulu-Natal province were sampled, 27 comprehensive interviews were conducted with 16 municipal officials, four environmental consultants (ECs), and seven environmental management inspectors (EMIs).

### Results

The findings suggest that institutional capacity, regulatory enforcement, and political support are crucial determinants of compliance. Furthermore, the study identifies awareness, capacity building, and organisational restructuring as pivotal strategies for enhancing environmental compliance.

### Conclusions

Improving municipal environmental compliance requires stronger leadership commitment, institutional capacity, consistent enforcement, and collaboration. The study contributes to environmental governance and sustainability by elucidating the role of institutional dynamics and political influence in shaping compliance. It provides a foundation for stakeholders, future research, and policy development aimed at effective environmental compliance.

**Data availability statement:** The data underlying the results presented in the study are available from https://doi.org/10.25399/UnisaData.30836537.

**Funding:** The author(s) received no specific funding for this work.

**Competing interests:** The authors have declared that no competing interests exist.

## Introduction

The attainment of environmental compliance is crucial for the promotion of sustainable development and the protection of natural resources. Although it is essential to ensure environmental compliance across all spheres of government in South Africa, it is particularly incumbent upon municipalities, which bear the responsibility for service delivery. Empirical evidence indicates that numerous municipalities face significant challenges in adhering to environmental legislation, primarily due to constraints such as limited resources [1], insufficient budget allocations [2], and political interference [3].

Over the preceding thirty years, South Africa has experienced significant advancements in environmental legislation, culminating in a comprehensive and intricate legal framework that regulates all aspects of environmental management [4]. Nonetheless, notwithstanding these legislative advancements, persistent non-compliance in municipalities indicate concerns regarding environmental governance effectiveness and accountability [5].

Environmental compliance can be defined as the adherence of individuals or organisations to environmental legislation, regulations and permit conditions established to safeguard the environment and advance sustainable resource utilisation [6]. From a governance perspective, it represents not only a regulatory requirement but also an indicator of accountability and stewardship [7]. It is integral to ensuring that anthropogenic activities do not result in the degradation of ecosystem goods and services. It facilitates the management and prevention of air, soil, and water pollution, waste generation, and climate change [6].

For environmental compliance to be effective, it requires both behavioural and economic transformations [7]. It represents a pivotal step toward achieving sustainable development, differentiating it from other forms of compliance. Sustainable development necessitates a balance between environmental imperatives and developmental objectives, requiring continuous inquiry and evaluation. Consequently, the imposition of fixed regulations, as observed in other types of legislative compliance, is problematic. Municipalities struggle with compliance to legislation due to fixed legislation and lack of adaptability. Fixed regulations often prove inadequate as they rely on existing knowledge, whereas flexible systems, although initially less optimal, are more prudent in the long term by addressing low-probability, high-impact events [8].

The body of literature regarding environmental compliance in South Africa is relatively sparse. Academic focus has predominantly been on the review of legislative frameworks, the role of municipalities in environmental governance, and private sector compliance, with limited consideration of how governance dynamics impact environmental compliance. This gap is significant, given that in South Africa, municipalities serve as both regulators and the regulated in environmental management. Their compliance capability is influenced by factors such as political dynamics [3], capacity building [9], communication [10], institutional frameworks [11], ignorance [12], and fragmented governance [1]. Identifying and addressing these determinants is crucial to promoting pro-environmental behaviour and ensuring compliance [7].

To understand and interpret the approaches of environmental governance used in municipalities and how they influence compliance, the study used the Governance Theory (GT).

## Theoretical underpinnings

GT can be used to elucidate the complex dynamics inherent in local environmental governance. Initially conceptualized by Rod Rhodes in 1997, GT provides an analytical lens through which the influence of governance structures and processes on policy outcomes and organizational behaviour can be examined [13,14]. This theoretical approach is critical in analysing how local governments administer and implement environmental legislation. GT underscores the impact of governance structures on policy execution, encompassing hierarchical models, networked approaches, and market-based mechanisms.

Hierarchical governance is characterized by a conventional model that operates through a top-down approach, wherein decisions are disseminated from higher levels of authority [14]. Within this paradigm, adherence to environmental legislation is dependent on the precision and enforcement of said legislation. Municipal officials are likely to comply with legislation when it is effectively communicated and overseen by leadership. However, the top-down model can sometimes result in diminished engagement or understanding.

In contrast, networked governance emphasizes collaborative efforts and coordination among diverse stakeholders [14]. This model encourages cooperation and the exchange of information. By engaging stakeholders and participating in collaborative endeavours, officials gain access to resources, expertise, and best practices that facilitate environmental compliance. Thus, governance significantly influences compliance within local government structures.

The study endeavours to investigate the determinants and strategies that influence environmental compliance within municipalities. By integrating insights from municipal officials, EMIs, and ECs, the research aims to bridge the existing gap between governance structures and compliance outcomes. Moreover, the study furnishes both short-term and long-term actionable recommendations to municipal managers, regulators, and policymakers.

## Methods

A qualitative research design, grounded in governance theory, was employed to investigate the determinants and strategies for environmental compliance within municipalities. Semi-structured interviews were conducted with municipal officials, ECs, and EMIs. The investigation was guided by the following research questions: What are the factors influencing compliance with environmental legislation? What are the perspectives of consultants and inspectors on municipal compliance? What interventions are required to enhance compliance in municipalities?

### Data collection, ethics, participants and sampling

Ethical clearance was obtained from the Research Ethics Committee of the College of Agriculture and Environmental Sciences of the University of South Africa (ethical clearance number: 2022/CAES_HREC/180). Furthermore, approval was obtained from each of the 10 selected municipalities. These municipalities were purposely selected based on a set of criteria including their diversity, historical compliance records, and specific governance attributes. The selection encompassed both municipalities with high and low performance metrics, as well as those with and without dedicated environmental units, to comprehensively encapsulate the variability in governance capacity and institutional responsiveness.

The interpretivist paradigm guided this study, accentuating knowledge acquisition through profound interpretation of phenomena [15]. This paradigm was deemed suitable for an in-depth understanding of the drivers of compliance in municipalities and the requisite interventions for improvement.

A qualitative approach facilitated a gradual understanding of these factors by employing data comparison and contrast. The case study research design facilitated a thorough exposition, comparison, and pattern identification. Purposeful sampling techniques were utilized to select all participants. A total of 27 participants (16 municipal officials, four ECs and

seven EMIs) were sampled, representing various key departments. The adequacy of the sample size was evaluated in accordance with the methodological guidelines posited by Bekele and Ago [16]. These guidelines emphasize the importance of accounting for factors such as sample composition, the scope of the study, the intrinsic nature of the research topic, the quality of collected data, and the research design. Furthermore, the process of iterative analysis was recognized as crucial in this assessment.

ECs chosen were those employed as environmental control officers on municipal projects, complemented by EMIs with relevant municipal experience. All participants were given information sheets regarding the study, and they provided written consent prior to their participation. Anonymization was used to protect research participants from the accidental breach of confidentiality. The names and identities of the participants were not revealed in the findings of the study. Participants were assigned numbers to conceal their identities. Recordings and transcripts were securely stored in password-protected folders accessible to only the researcher.

Semi-structured interviews were conducted from March 2023 to June 2024, commencing with the collection of demographic data from participants. The Interviews followed a guide (Annexure A) based on research objectives and the study's theoretical framework, featuring open-ended questions to gain deep insights into participants' perceptions. The study was piloted with two participants outside the main sample to ensure clarity and alignment. Feedback was used to refine the questions and flow for rich, reflective responses.

Increased engagement through additional interviews enabled a more profound exploration and probing of these factors. The adaptability of the semi-structured interviews facilitated participant engagement while fostering an environment of trust and openness. Each interview was conducted over an approximate duration of 60 minutes.

## Data analysis

Data were analysed per the foundational principles of qualitative data analysis, which encompass concurrent data collection and analysis, iterative analysis, and the constant comparative examination of data [17]. We followed a six-phase process described by Braun and Clarke [18] in analysing the data. A substantial amount of time was dedicated to the transcription of the data. The process began with an intensive review of each transcript to comprehend the content. This involved meticulous examination, identification, and coding of significant text segments. During this phase, words were scrutinized individually to discern conceptual themes.

The process resulted in 513 codes. Identically coded segments were aggregated into categories, enabling the visualization and identification of patterns and relationships among categories, ultimately culminating in the identification of three principal themes. To enhance the trustworthiness of the analysis, we used measures and accepted procedures, which included credibility, transferability, dependability, and conformability. These included coding reliability, triangulation, member checking, audit trial, and transferability. During the analytical phase, we engaged in peer debriefing sessions to rigorously evaluate emergent themes, scrutinize underlying assumptions, and maintain interpretive transparency.

To complement the qualitative data analysis and enhance the validity of the themes, a quantitative analysis of the interview data was also undertaken. Coding frequencies and co-occurrences were quantified to evaluate the relative emphasis of identified constructs such as enforcement, political support and resource availability. These frequencies and co-occurrences were not subjected to inferential statistics but were used to contribute to internal generalisability of the qualitative findings. The integration of quantitative content analysis in a qualitative design aligns with recommendations by Maxwell [19] to allow for the presentation of evidence for the interpretations and improve data transparency.

## Results

Our qualitative analysis yielded thematic outcomes that elucidate the determinants and strategies necessary to enhance compliance within municipalities. These themes furnish critical insights from municipal officials and key stakeholders within local government.

## Environmental literacy, a robust environmental unit, and enforcement significantly influence compliance in municipalities

Environmental literacy, the presence of a robust environmental unit, and enforcement were identified as influential factors affecting compliance levels. The predominance of these themes suggests an inadvertent neglect of other potential compliance factors. These issues were prominently featured in discussions regarding the drivers of municipal compliance. A considerable number of participants highlighted environmental literacy as the principal factor impacting compliance.

For instance, Participant 8, aged 57 an EC by profession, offered a comprehensive perspective by identifying environmental literacy as a pivotal compliance driver: "*I think it depends on literacy at the levels that matter. If you have someone strong who is environmentally literate, you are likely to succeed and see some of the plans considering the environment. So, for me, what is critical is that at a political level because these people are decision makers; your mayors, councillors, if we can try and infuse this thinking of the environment in a way or even force it in a particular way. By forcing, I am not saying we request for a qualification, but I am saying maybe before you become a councillor you must attend the induction where there is a component of environmental awareness.*

*Furthermore, within the existing Councillors Induction Programme, it is imperative to include some form of assessment. Observations from attending these inductions reveal that councillors often engage in distractions, such as phone calls, indicating a lack of seriousness toward the program. Implementing assessments would compel thought and engagement, ensuring the material is considered. While individuals can compile portfolios of evidence, mandating reflective assessments could effectively cultivate critical thinking. It is not suggested that a master's degree in environmental science is necessary, but thought-provoking initiatives should be incorporated.*"

Consistent regulatory enforcement was also underscored as a pivotal motivator for municipal compliance. According to Participant 20, aged 34, enforcement serves as a critical compliance trigger: "*Both administrative and criminal enforcement represent key compliance drivers for municipalities and other organisations alike.*"

The findings indicate that environmental literacy, a robust environmental unit, and enforcement are pivotal determinants influencing compliance within municipalities. Nevertheless, ECs and EMIs discern a prevalent reluctance to comply among municipalities.

## ECs and EMIs perceive municipalities as lacking willingness to comply with environmental legislation

Data analysis corroborates that EMIs, and ECs consistently perceive an absence of compliance willingness in municipalities. Participant 15 provides further elucidation: "*I am uncertain whether it stems from a lack of comprehension or an intentional refusal to comply, yet I incline towards the perspective of potential misunderstanding or outright non-compliance.*"

The data analysis reveals that municipalities are cognizant of environmental requirements, yet they fail to adhere to the stipulations of environmental permits. This is exemplified by Participant 22, aged 35, who states: "*Based on my experience, it is evident that they are aware of the requirements, as they apply primarily to advance their developments. However, upon obtaining authorisations, there is a tendency to overlook the necessity of adhering to the conditions associated with these approvals or authorisations.*"

Additionally, municipalities often conceal non-compliance, particularly in remote regions, as indicated by Participant 23: "*In certain instances, projects situated near towns and townships seek authorisation, yet those located in inaccessible rural and remote areas are developed unlawfully. Thus, compliance is maintained in visible areas, but not in those unseen.*"

The data analysis underscores that ECs and EMIs view municipalities as generally unwilling to comply with environmental legislation. While some perceive a lack of understanding, others have noted significant non-compliance in projects situated in remote areas.

## Awareness, capacity building, organisational restructuring, and political support improve compliance in municipalities

The study suggests that awareness, capacity building, organisational restructuring, and political support are instrumental in enhancing compliance within municipalities. It is posited that awareness and capacity building are pivotal in improving compliance. For instance, Participant 3 suggests: "*A viable solution would be for the department to institute a programme aimed at training and increasing the awareness of councillors, particularly municipal executive officers, to sensitise them to the significance of environmental management, as this could initiate the process.*"

The data further indicates a requirement for organizational restructuring to enhance compliance within municipalities. The following comment succinctly encapsulate this necessity: "*It is recommended that municipalities establish an internal unit dedicated to environmental compliance. Such a unit must be conversant with legal requirements. An internal advocate would facilitate coordination across various municipal sectors, as opposed to relying on provincial authorities dealing with broader jurisdictional challenges. Following its establishment, there could be opportunities for collaborative efforts*" (Participant 20).

One participant indicated that the qualified person should be appointed at a senior management level. "*Appointing a junior individual may not command the necessary respect, hence it is crucial to have a strong representative capable of engaging with senior officials and politicians*" (Participant 8).

Political support has been identified as a key intervention to enhance compliance within municipalities. Participant 24 provides an illustrative example that underline the significance of leadership in environmental compliance. "*Decision-makers in positions of power, particularly in local municipalities hold regulatory mandates. Their perception of environmental significance is critical, as some may prioritize infrastructural projects over environmental concerns, leading to suboptimal long-term outcomes, such as infrastructure failure. The understanding and valuation of environmental priorities by those in power are essential. While political appointments are prevalent in local municipalities, there is a necessity for appointments based on specialisation, qualifications, and skills. These individuals should drive compliance, yet they often operate under the directives of their superiors, complicating their responsibilities*" (Participant 24).

The data analysis indicates that several strategies can be instituted within municipalities to enhance compliance. Specifically, awareness, capacity building, organizational restructuring, and political support have emerged as the principal measures necessary for improving environmental compliance in municipalities.

## Quantitative analysis of interviews

The quantitative analysis of compliance drivers identified that EMIs predominantly highlighted enforcement as the principal driver of compliance, reflecting a predilection for a traditional compliance approach, rooted in command-and-control-based governance mechanisms. Conversely, ECs underscored the importance of environmental literacy and stakeholder support, advocating for a participatory and knowledge-based strategy. Furthermore, both groups acknowledged the significance of political backing and resource availability as compliance drivers within municipalities (Fig 1).

This figure summarises the occurrence frequency of the key drivers of compliance identified through interview coding. The visual depicts patterns amongst high and low performing municipalities.

The analysis of code co-occurrence demonstrated a significant correlation between the comprehension of environmental issues and political support (Fig 2), suggesting that municipal leaders possessing an understanding of environmental concerns are inclined to promote compliance. Environmental literacy was also discussed by participants in conjunction with the availability of resources and the involvement of stakeholders.

The quantitative evaluation of principal interventions across all sectors indicated that the presence of qualified personnel, capacity building, and political support constituted the primary interventions necessary for enhancing environmental compliance in municipalities (Fig 3). ECs additionally stressed the importance of collaboration and governance

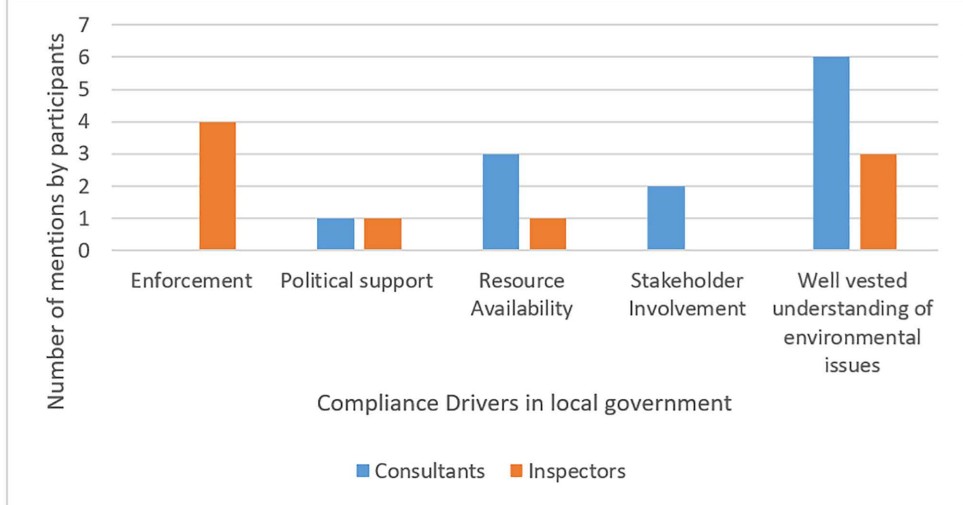

**Fig 1. Distribution of environmental compliance drivers identified across municipalities.**

enhancement, implying a demand for enhanced inter-stakeholder engagement. Inspectors concentrated on planning, budget allocation, and transformations in political administration. In contrast, municipal officials emphasized the importance of awareness, civic involvement, and organisational restructuring, indicating a focus on internal development.

## Discussion

This study identifies that environmental literacy, the incorporation of environmental units, and the implementation of regulatory enforcement actions exert a substantial influence on compliance levels within municipalities. This finding corroborates the institutional theory's assertion that institutional structures and norms exert a profound impact on organisational behaviour. The finding underscores the critical role of both internal and external factors in achieving compliance. Consequently, these results contribute to the advancement of understanding regarding the influence of institutional factors on compliance.

Empirical data provide evidence of the pivotal role that environmental literacy plays in ensuring adherence to environmental legislation in municipalities. This perspective is corroborated by Ardoin et al. [20], who highlighted the substantial role of environmental education in promoting sustainability. Furthermore, the presence of a robust environmental unit is instrumental in driving compliance within municipalities. When these units are well-resourced and effectively managed, their success in ensuring compliance is enhanced. These observations are consistent with the study by Pasquini and Shearing [5], which emphasized the importance of institutional strength in ensuring adherence to pertinent legislation within municipalities. Nevertheless, as noted by Mirumachi and van Wyk [21] and Sancho [22], well-resourced units may confront challenges in the absence of support.

The study's findings indicate that enforcement significantly influences adherence to environmental legislation. The finding that enforcement secures compliance is crucial, as successful environmental compliance constitutes a vital component of good governance, characterized by open, participatory, accountable, and transparent processes. Short [23] showed the role of enforcement in reinforcing adherence.

The finding indicates that compliance in South African municipalities is mainly achieved through hierarchical governance approach. Although this is in line with the data presented by other scholars such as by Lo et al. [24] illustrating the impact of enforcement mechanisms on compliance. Modern international environmental governance

**Fig 2. Code co-occurrence analysis revealing patterns and connections of factors driving environmental compliance in municipalities.**

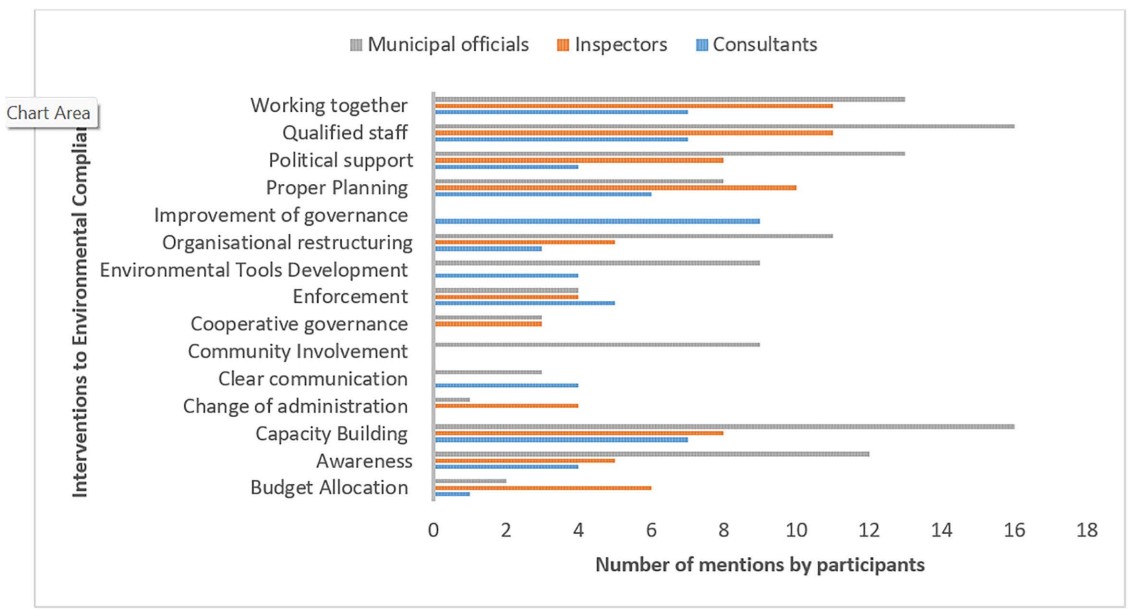

**Fig 3. Distribution of strategies to enhance environmental compliance behaviour suggested by participants across municipalities.**

advocate for co-operative approaches to compliance. For instance, Carter [25] reported that countries such as China in last few decades, underwent a transition from a command- and control-based approach with fixed targets towards a more open management style that integrate various stakeholder groups and promote sustainable economic development.

The study's findings further emphasize the need for an integrated approach to environmental governance. Enhancing environmental literacy ensures an informed and engaged community, strengthening environmental units augments the capacity to manage and comply with legislation, and enforcement guarantees adherence. Collectively, these elements engender a synergistic effect that enhances overall compliance and environmental performance.

This finding is consistent with recent literature that underscores the significance of multiple factors in attaining compliance [11] including environmental literacy and robust institutional frameworks. These factors are interdependent and exert a combined impact on ensuring compliance.

Data from the perspectives of consultants and inspectors suggest that these stakeholders perceive municipalities as resistant to adhering to environmental legislation. This finding carries several implications for policymaking and is situated within a broader research context on environmental management. The perception of non-compliance suggests a possible disjunction between the expectations of consultants and inspectors and the actual practices within municipalities. This discrepancy can be influenced by a variety of factors, including disparate understandings of what constitutes compliance, and the challenges municipalities encounter in achieving compliance.

The perception of resistance may be shaped by perceived systemic barriers, such as resource constraints or the absence of clear regulatory guidance. This finding is congruent with compliance theory. According to this theoretical framework, a municipality's willingness to comply is influenced by its perception of the legitimacy and feasibility of environmental legislation. The finding also corresponds with the study by Healy [26], which emphasized that municipalities often face competing priorities, such as economic development and immediate political pressures, which can overshadow environmental considerations. This suggests that what may be perceived as a lack of willingness may reflect higher-priority challenges confronting municipalities.

Scholars have noted that various challenges within local government influence their willingness to comply with legislation. Du Plessis [27] reported that legitimacy and trust are pivotal in fostering compliance. Improving environmental literacy ensures an informed and engaged community, strengthening environmental units enhances the capacity to manage and comply with legislation, and enforcement ensures adherence. Together, these elements create a collective effect that enhances overall compliance and environmental performance.

The data suggests that enhancing compliance in South African municipalities necessitates interventions such as awareness, capacity building, organisational restructuring, and political support. This finding is vital for the attainment of environmental compliance within municipal frameworks. The data reveals that environmental compliance transcends mere adherence or technical implementation, instead fostering an environment where compliance becomes sustainable. The data suggests that compliance requires understanding and acknowledging the significance of environmental legislation within the municipality through awareness. Nevertheless, it was discovered that awareness alone is insufficient; it is imperative to translate awareness into practical action via capacity building. Moreover, the data indicates that achieving compliance frequently demands organisational restructuring aimed at incorporating environmental management and enhancing coordination, communication, accountability, and effectiveness.

The participants advocated that political support is crucial to render these efforts towards compliance feasible. This implies that robust leadership and dedication at the political level are imperative to galvanize compliance within municipalities. The data offers lucid strategic direction for municipal managers and policymakers to consider in their efforts to achieve compliance. It reflects the necessity of fostering awareness and capacity regarding legislation, while also contemplating requisite organisational modifications and striving for substantial political backing. Ensuring that compliance is achieved and sustained within local government is crucial. The findings therefore supports GT's hierarchical governance framework in municipalities where the endorsement of the importance of compliance is driven by the senior management and implemented by the administrative staff.

This finding is corroborated by existing literature and aligns with recent studies [28,29] on environmental governance. Literature substantiates the significance of capacity building for achieving environmental compliance. For instance, Sherwood [30] emphasized the importance of organisational and institutional learning for capacity building and effective environmental management. The requirement for organisational restructuring is further supported by the studies of Pasquini and Shearing [5]. Ying et al. [31] found that organisational design bears significant implications for sustainability.

Notwithstanding, the literature also highlights potential challenges, especially concerning capacity building. For instance, Sherwood [30] elucidated that while capacity building is widely recognised as indispensable, its efficacy is often constrained by resource limitations. Endale et al. [32] identified cultural differences as barriers to the implementation of capacity building. Kulesa et al. [33] revealed how cultural tensions negatively influence capacity building.

The data underscores the critical importance of political will as a foundational element for all subsequent efforts. In their absence, programmes aimed at raising awareness, initiatives for capacity building, or organisational restructurings may fail to achieve the intended impact. Political support thereby emerges not merely as a component, but important in realizing environmental compliance within local government in South Africa. The substantial role of political support in environmental governance is well-documented in the literature [1,3,34]. Pasquini et al. [35] indicated that securing political support exerts a positive influence on environmental governance. Thakur and Nel [3] demonstrated that politics is intrinsically linked to environmental governance, particularly in the realm of planning. Switzer [36] also found that municipalities in the United States are often constrained in their environmental decision-making by state politics.

Nevertheless, although the reliance on political support is crucial, it is prone to instability and fluctuations accompanying electoral cycles, thereby posing a threat to sustained compliance efforts. The data denote a connection among these factors within municipalities, with some requiring resolution before others can achieve effectiveness. This interlinked pattern of factors suggests that initiatives in one domain can magnify impacts in others, instigating a cascading effect that facilitates more effective compliance.

Achieving environmental compliance can therefore be conceptualised as a function of three governance determinants: institutional capacity, regulatory enforcement and political support (Fig 4). These factors are important for enhancing compliance in municipalities. Interventions such as awareness, capacity building and organisational restructuring are posited as strategies to sustain compliance and promote institutional accountability. The proposed model suggests that outcomes such as improved future governance performance may be realised when these determinants and strategies are effectively implemented.

The findings suggest that environmental compliance within municipalities is a complex issue, shaped by both individual and institutional dynamics. They contribute to and expand the framework of GT. Good governance is characterized by its focus on key elements, namely accountability, participation, transparency, and responsiveness [37]. The EMI's emphasis on the significance of enforcement as a compliance driver reveals a governance model grounded in regulatory accountability. Their perspective suggests that inadequate enforcement constrains compliance, underscoring the necessity for robust oversight systems within governance frameworks. The prominence of resource availability, political support, and

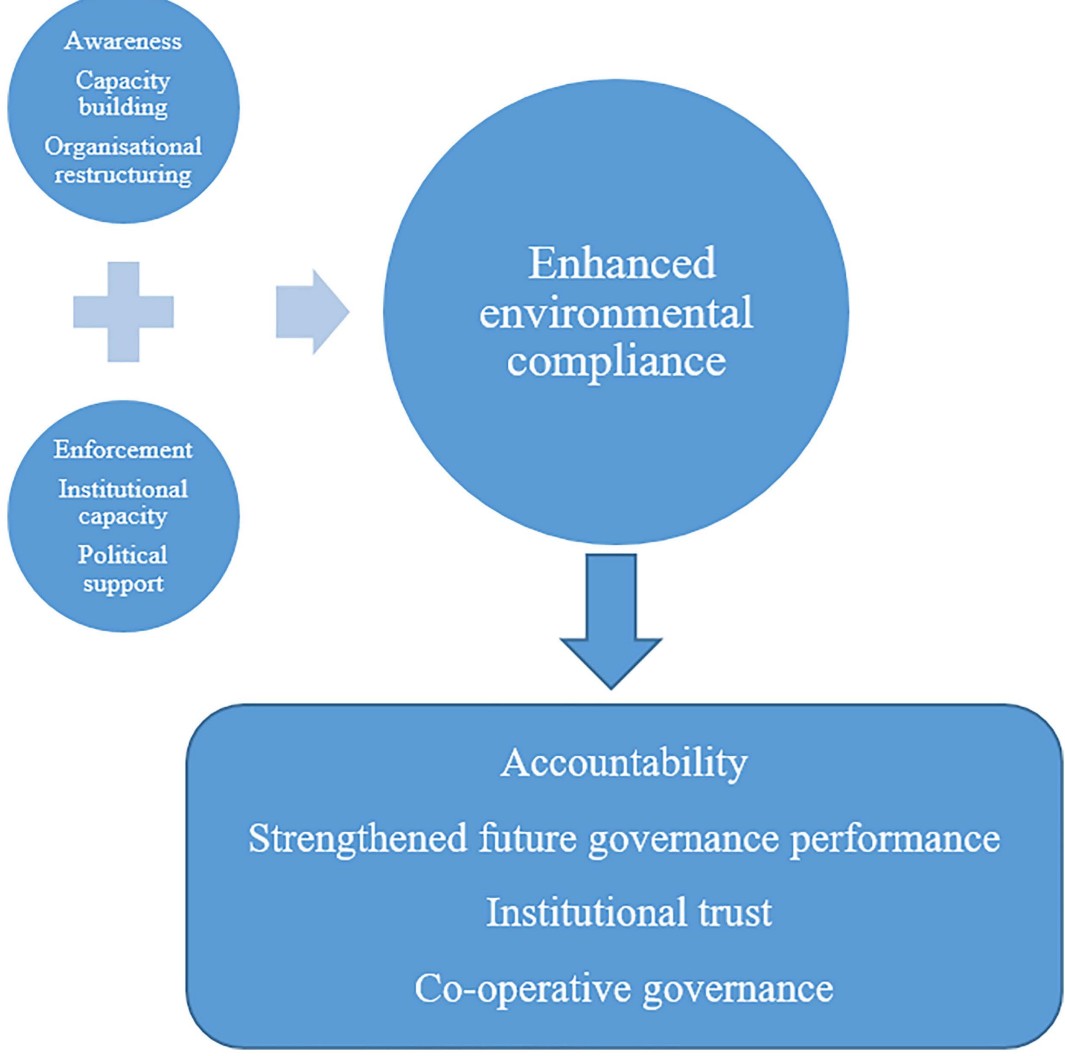

**Fig 4. Conceptual model linking governance determinants, strategies and compliance outcomes.**

capacity building among all participants underscores the importance of institutional capacity in governance. Conversely, the municipal official's concentration on internal mechanisms, such as organizational restructuring, highlights the significance of internal coordination for sustainable development [7].

The EC's focus on effective communication and stakeholder engagement reflects the essential nature of transparent and participatory governance. Other researchers [38] have similarly recognized these soft governance structures as critical in fostering trust and legitimacy within governance. Our study's recognition of political support and administrative changes underscores the critical role of political will and leadership stability in ensuring adherence to environmental legislation.

The study also unveiled divergent perspectives, with the EMI advocating for control and oversight, the EC emphasizing systems thinking, and the municipal official advancing internal development as strategies to secure compliance. These disparate perspectives underscore the distinct roles occupied by each group within the governance network implementers, advisors, and regulators highlighting the necessity for integrated, multi-actor governance strategies.

## Conclusions

The article evaluated the determinants and strategies essential for enhancing compliance within municipalities. The findings indicate that municipalities with robust environmental units and receive regulatory enforcement actions are more adept at fulfilling compliance standards, suggesting that investment in these domains can substantially enhance environmental compliance and sustainability.

Awareness, capacity building, organisational restructuring and political support emerged as the four strategies to improving and sustaining compliance in municipalities. GT provided lens to understand how these factors interact. Political support was identified as pivotal in influencing compliance, while a capacitated environmental unit ensured implementation of legislation.

The study recommends a set of interventions to enhance compliance in South African municipalities. In the short term, municipalities should focus on improving awareness of environmental compliance, appointing qualified environmental persons, and enhancing collaboration with stakeholders. Regulators should increase enforcement visibility and offer technical guidance. Policymakers should support capacity building and incorporate compliance performance into monitoring systems.

Long term, municipalities should establish dedicated environmental units, integrate compliance in planning and budgeting, and foster a proactive compliance culture. Regulators should pursue collaborative compliance partnerships, while policymakers ensure ongoing political and financial support for environmental governance. Aligning capacity, enforcement, and commitment will help municipalities contribute to sustainability and meet their constitutional duties. This aligns with the broader understanding that a supportive culture, comprehensive training, and adequate resources are imperative for effective environmental management.

The findings of this study underscore the importance of multilevel interventions that fortify capacity, foster accountability, and bolster stakeholder support. Achieving environmental compliance necessitates not only legislation and resources but also accountable leadership, transparent communication, and shared responsibility in municipal governance. Future studies should expand on these insights through applying comparative approaches to investigate linkages between governance and compliance across different contexts.

### Limitations

The study draws perceptions and perspectives from municipal officials, ECs and EMIs. The study involved small, non-random samples meaning the findings are context specific to South African municipalities and may not be generalizable. However, the analytical insights are transferable to similar municipalities as those sampled. Future studies could benefit from a mixed methods approach to provide a more comprehensive understanding.

## Supporting information

**S1 File. Annexure A.**
(PDF)

## Acknowledgments

This article is based on the author's thesis entitled 'Perceptions of environmental compliance and pro-environmental behaviours in KwaZulu-Natal municipalities, South Africa' towards the Doctor of Philosophy degree in Environmental Management at the School of Agriculture and Environmental Sciences of the University of South Africa, South Africa, in February 2025, with Gerhard P. Nortje, the supervisor.

## Author contributions

**Conceptualization:** Nqobile S. Zungu.

**Data curation:** Nqobile S. Zungu.

**Formal analysis:** Nqobile S. Zungu.

**Investigation:** Nqobile S. Zungu.

**Methodology:** Nqobile S. Zungu.

**Project administration:** Nqobile S. Zungu.

**Resources:** Nqobile S. Zungu.

**Software:** Nqobile S. Zungu.

**Supervision:** Gerhard P. Nortjé.

**Visualization:** Nqobile S. Zungu, Gerhard P. Nortjé.

**Writing – original draft:** Nqobile S. Zungu.

**Writing – review & editing:** Gerhard P. Nortjé.

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
