## [Decision Letter · Decision Letter 0]

23 Oct 2025

Dear Dr. Zungu,

Thank you for submitting your manuscript to PLOS ONE. After careful consideration, we feel that it has merit but does not fully meet PLOS ONE’s publication criteria as it currently stands. Therefore, we invite you to submit a revised version of the manuscript that addresses the points raised during the review process.

We look forward to receiving your revised manuscript.

Kind regards,

Muhammad Luqman

Academic Editor

PLOS ONE

Journal Requirements:

**Additional Editor Comments:**

I have read your Mnuscript in detaild and suggested a amjor revison. My detailed comments can be found as follows:

1. Clarify research gap and originality compared to prior studies.

2. Strengthen theoretical framework linking governance theory to findings.

3. Provide justification for sample size of 27 participants.

4. Explain how municipalities were purposively selected.

5. Include more detail on interview structure and questions.

6. Discuss data reliability and potential researcher bias.

7. Improve consistency in referencing recent South African studies.

8. Expand on limitations and implications for generalizability.

9. Add a conceptual model connecting determinants and strategies.

10. Quantitative analysis section lacks statistical clarity—revise.

11Figures are underexplained; improve captions and visual quality.

12. Integrate discussion with international environmental governance literature.

13. Ensure results and discussion are clearly differentiated.

14. Strengthen link between findings and policy recommendations.

15. Revise abstract for brevity and focus on main contributions.

16. Clarify ethical considerations regarding participant anonymity.

17. Improve coherence between introduction and conclusion claims.

18. Justify inclusion of quantitative content analysis in a qualitative design.

19. Provide citation support for claims about political influence.

20. Revise for grammatical consistency and eliminate redundancy.

Reviewers' comments:

Reviewer's Responses to Questions

**Comments to the Author**

1. Is the manuscript technically sound, and do the data support the conclusions?

Reviewer #1: Yes

2. Has the statistical analysis been performed appropriately and rigorously?

Reviewer #1: Yes

3. Have the authors made all data underlying the findings in their manuscript fully available?

Reviewer #1: Yes

4. Is the manuscript presented in an intelligible fashion and written in standard English?

Reviewer #1: Yes

Reviewer #1: Reviewer Comments for the Authors

1. Title and Abstract

• The abstract provides a concise summary, but would benefit from clearly separated sections (e.g., Background, Methods, Results, Conclusion) to improve reader accessibility and conform with best practice. I would suggest the authors to reformat the abstract into structured subheadings to enhance clarity.

2. Introduction

• However, lines 49–56 should be restructured to clarify the distinction between compliance vs. enforcement and the challenge of “fixed rules” in sustainability. I would suggest that the authors to briefly define environmental compliance versus enforcement and highlight examples where legislation may fall short due to institutional fragmentation.

3. Theoretical Framework

• The integration of governance theory (GT) is appropriate and enhances the conceptual depth of the study.

• However, the connection between GT and the study’s coding themes could be strengthened in the discussion. The authors should explicitly link themes like political support, enforcement, and institutional restructuring to GT’s dimensions (e.g., hierarchical, networked, market-based governance) in the discussion section.

4. Methodology

• Ethical protocols, data collection, and participant demographics are clearly presented.

• The summary of methods lacks sufficient detail on:

o Interview guide development,

o Justification for the sample size (n = 27), and

o Coding reliability/validation measures.

The authors should include a supplementary file with the semi-structured interview guide. Clarify whether inter-coder agreement or peer debriefing was used to validate themes.

5. Results

• Results are rich and supported with meaningful quotes.

• The inclusion of Figures 1–3 is commendable, especially:

o Figure 1 (page 34): Compliance drivers by role group,

o Figure 2 (page 35): Co-occurrence of drivers, and

o Figure 3 (page 36): Preferred interventions by stakeholder group.

• However, the narrative tends to over-quote participants. Several quotes repeat the same theme.

Focus on integrating fewer but more representative quotes and enhance comparative analysis between participant categories (e.g., ECs vs EMIs).

6. Discussion

• The discussion aligns well with findings and relevant literature.

• Some claims are too general (e.g., “political will is a cornerstone…”), and citations need to be used more selectively.

• The link between findings and institutional theory could be sharpened, as this would strengthen the study’s theoretical contribution.

Provide a schematic or conceptual figure showing how identified themes map onto GT dimensions. This would help readers visualize how governance structures shape compliance.

7. Conclusion and Implications

• The conclusion is policy-relevant and forward-looking.

• However, the recommendations are broad; prioritization would enhance their practical value. I would recommend specific actions tailored to different stakeholders (e.g., municipal managers, regulators, policymakers) and identify short- vs long-term priorities.

8. Formatting, Clarity & Language

• Some editorial polishing is needed:

o Overuse of passive voice,

o Redundant wording in certain sections,

o Occasional lapses in sentence clarity (e.g., lines 253–259).

The paper minor language and grammar editing, especially to improve sentence flow and eliminate verbosity.

**Do you want your identity to be public for this peer review?** For information about this choice, including consent withdrawal, please see our Privacy Policy

Reviewer #1: No

---

## [Author Response · Author response to Decision Letter 1]

10 Nov 2025

We sincerely thank the Editor and Reviewers for their constructive comments and for recognizing the potential contribution of our manuscript to PLOS ONE. We have carefully considered each point raised and made substantial revisions to enhance the clarity, rigor, and theoretical coherence of the paper. In the response to the reviewers document we have provided a detailed, point-by-point response indicating the changes made.

---

## [Editor Report · Decision Letter 1]

26 Nov 2025

Determinants and strategies for environmental compliance in municipalities: Perspectives from KwaZulu-Natal Province, South Africa

PONE-D-25-23589R1

Dear Dr. Zungu,

We’re pleased to inform you that your manuscript has been judged scientifically suitable for publication and will be formally accepted for publication once it meets all outstanding technical requirements.

Kind regards,

Muhammad Luqman

Academic Editor

PLOS ONE
---

## [Editor Report · Acceptance letter]

PONE-D-25-23589R1

PLOS One

Dear Dr. Zungu,

I'm pleased to inform you that your manuscript has been deemed suitable for publication in PLOS One. Congratulations! Your manuscript is now being handed over to our production team.

Kind regards,

on behalf of

Dr. Muhammad Luqman

Academic Editor

PLOS One